# A virtual screening and molecular dynamics approach in search of novel antibiotic chemotypes

Tahira Noor[1,2,3], Daniel C. Schultz[2], Yuting Zhai[4,5], Hannah E. Snoke[6], Suyeun Noh[4,5], Gustavo Seabra[2], Richard E. Lee[6], Kwangcheol Casey Jeong[4,5], Chenglong Li[2], Abdul Rauf Siddiqi[1]*

1 Department of Biosciences, COMSATS University Islamabad (CUI), Islamabad, Pakistan, 2 Department of Medicinal Chemistry, College of Pharmacy, The University of Florida, Gainesville, Florida, United States of America, 3 Department of Bioinformatics, International Islamic University Islamabad (IIUI), Islamabad, Pakistan, 4 Emerging Pathogens Institute, Department of Animal Sciences, University of Florida, Gainesville, Florida, United States of America, 5 Department of Animal Sciences, University of Florida, Gainesville, Florida, United States of America, 6 Department of Chemical Biology & Therapeutics, St. Jude Children's Research Hospital, Memphis, Tennessee, United States of America

* araufsiddiqi@comsats.edu.pk

## Abstract

Due to the constantly evolving threat of antibiotic resistance, there is a dire need for novel antibacterial agents. Dihydropteroate synthase (DHPS) is a key bacterial enzyme which has been targeted for nearly a century as a means of selective treatment of microbial infections and exhibits two orthosteric binding sites – the *p*-aminobenzoic acid (*p*ABA) site and the pterin site. The former is the target of sulfonamides, the earliest class of synthetic antibiotics, and its mutant forms have conferred resistance to this drug class, diminishing its utility in the clinic. Conversely, the pterin site, which is highly conserved across bacterial species, is purported to be less tolerant of mutations, rendering it an attractive target for novel antibiotics. Inspired by this, we conducted a large virtual screen of more than 450,000 compounds from commercial databases, identifying compounds **8802** and **7034** as potential pterin-site inhibitors. Compound **8802** was quite attractive as a hit due to the ease of generating analogues, leading to the synthesis of novel compounds **LST-1** and **LST-2**. Rigid docking and molecular dynamics suggested favorable binding of these compounds to the pterin site of DHPS, and compound **8802** exhibited superior antibacterial activity compared to its analogues and **7034**. Fluorescence polarization assays did not indicate competitive inhibition of pterin-derived probe binding, and surface plasmon resonance (SPR) suggested these compounds bind very weakly to DHPS, in a nonspecific manner. The *in silico* assessment of the physicochemical and pharmacological properties predicted a favorable overall profile, indicating that these are suitable leads for further study to improve their activity and determine their precise mode of action.

**Data availability statement:** All relevant data are within the paper and its Supporting information files.

**Funding:** Ms. Tahira Noor was awarded a one-year secondment under the supervision of Dr. Chenglong Li at the University of Florida, USA from Higher Education Commission (HEC)Pakistan. The details of grant number are given below. FDP Grant # 17-5/FBSI-002/HEC/Sch-FDP/2018. The URL is https://www.hec.gov.pk/english/services/faculty/fdp/Pages/Financial-Benefits.aspx HEC has played no role in the study design, data collection and analysis, decision to publish, or preparation of the manuscript.

**Competing interests:** The authors have declared that no competing interests exist.

## Introduction

Treatment-resistant bacterial infections are a leading cause of death worldwide and present a growing public health threat. In 2021, bacterial antimicrobial resistance played a role in an estimated 4.71 million deaths globally, with approximately 1.14 million deaths directly caused by antibiotic-resistant infections, and by 2050 it is estimated that antimicrobial resistance will play a role in or be the direct cause of 8.22 million and 1.91 million deaths, respectively, worldwide [1]. Various antibiotic classes, including β-lactams, fluoroquinolones, aminoglycosides, macrolides, and glycopeptides, have been instrumental in saving numerous lives, however, the rapid development of bacterial antimicrobial resistance has limited their clinical efficacy [2]. Furthermore, from 2013 through 2022, only 19 small molecule drugs received approval for clinical use as antibacterial agents, none of which were first-in-class, highlighting an unmet need for antibiotics with novel mechanisms of action [3].

One classical target for antibacterial agents is dihydropteroate synthase (DHPS), a critical enzyme in the folate synthesis pathway of prokaryotes and primitive eukaryotes [4]. This enzyme has two binding pockets: one which binds 6-hydroxymethyl-7,8-dihydropterin pyrophosphate (DHPP, **1**) and one which binds *p*-amino benzoic acid (*p*ABA, **2**) [5]. DHPS facilitates the reaction of *p*ABA with DHPP to form 7,8-dihydropteroate **3** and pyrophosphate **4** through an $S_N1$-type mechanism, with 7,8-dihydropteroate being a crucial intermediate in folate biosynthesis (Fig 1) [6]. The blockade of DHPS activity in bacteria inhibits their growth and, since mammals lack DHPS and instead acquire folate through their diet, targeting DHPS offers an inherently selective means of treating bacterial infection [4,7].

Since the 1930s, members of the sulfonamide class of DHPS inhibitors have been used as *p*ABA-competitive inhibitors for bacterial infections [4]. Selected notable sulfonamide antibiotics include sulfanilamide **5**, sulfadiazine **6**, and sulfamethoxazole **7**, which are shown in Fig 2 A [7,8]. Due to the extensive clinical usage of sulfonamides, however, bacterial populations swiftly built up resistance to this class of antibiotics. To combat this bacterial resistance pathway, one promising strategy is to target an alternative binding site of DHPS – the pterin site. Unlike the *p*ABA site, which is located in the flexible loop region, the pterin pocket is embedded within the TIM barrel and is characterized by its rigidity and the presence of highly-conserved residues, as evidenced by crystal structures of DHPS from multiple organisms [9]. Exploiting these features of the pterin pocket, which are crucial for substrate specificity, could be advantageous as mutations that would confer resistance to inhibitors targeting this site would likely impair the enzyme's activity, making such resistance less likely to develop [9,10]. Furthermore, the conserved nature of the pterin pocket suggests that a well-designed inhibitor targeting this site could have broad-spectrum antibacterial activity and demonstrate efficacy against sulfa drug-resistant organisms. Pterin-site inhibitors generally mimic the heterocyclic core of DHPP, with their minimum pharmacophore consisting of a 2-aminopyrimidin-4(3*H*)-one and a hydrogen bond acceptor *para* to the amine group [9]. Since the 1940s, synthetic pterins have been studied as potential alternatives to sulfonamides, selected examples of which are shown in Fig 2B [11–16]. Joining in the renewed interest in antibiotics targeting the pterin

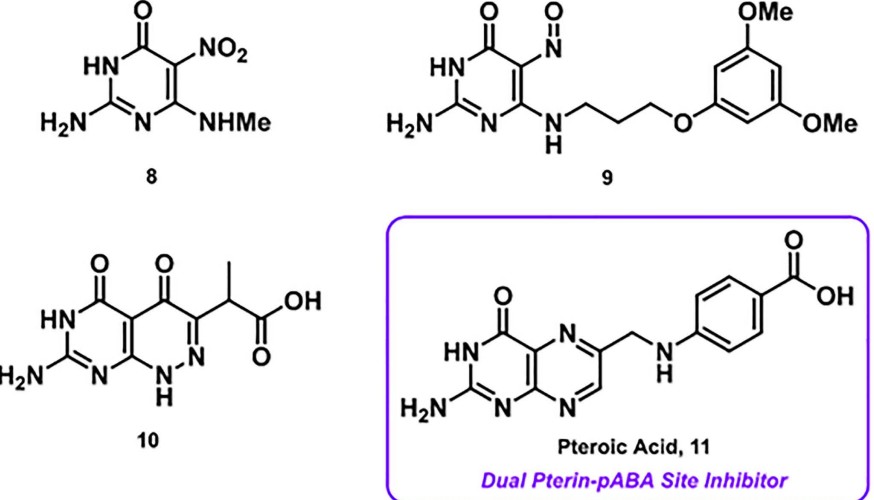

**Fig 1. DHPS-catalyzed condensation of DHPP and pABA yields 7,8-dihydropteroate and pyrophosphate.**

### A. Notable Sulfonamide Antibiotics Targeting the pABA Site of DHPS

Sulfanilamide, 5 Sulfadiazine, 6 Sulfamethoxazole, 7

### B. DHPS Inhibitors Targeting the Pterin Binding Site

8 9

10 Pteroic Acid, 11
*Dual Pterin-pABA Site Inhibitor*

**Fig 2. Structures of selected known inhibitors of DHPS.** A.) Key sulfonamide antibiotics targeting the pABA binding site of DHPS; B.) Notable pterin-based inhibitors from literature.

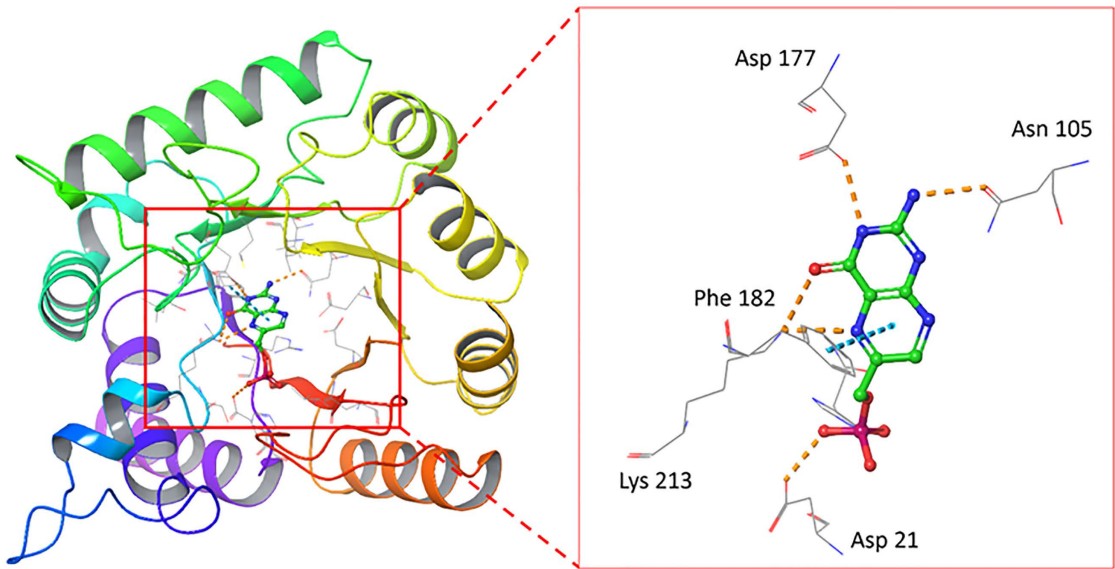

site within the last decade [17–20], the present study discloses our use of computer-aided drug design in pursuit of novel pterin-site inhibitors with activity against *Pseudomonas aeruginosa*, *Staphylococcus aureus*, *Escherichia coli*, and *Bacillus subtilis*. Resistant strains of *P. aeruginosa* and *S. aureus*, in particular, have been cited as high-priority pathogens against which novel antibiotics are needed [21].(Tacconelli et al., 2018).

## Results and discussion

### Virtual screening

Structure-based virtual screening is a viable and inexpensive alternative to high-throughput screening for the identification of novel bioactive hit compounds [22]. (Lionta et al., 2014). This method models the interaction of ligands with the target protein of interest to predict binding activity and can treat the receptor in either a rigid or flexible manner [22–24].

To investigate novel pterin-site inhibitors of DHPS, we used QuickVina 2.1 [25](Alhossary et al., 2015) to screen a database with >450,000 compounds from different commercial vendors against the high-resolution (1.7 Å) crystal structure of known pterin 6-hydroxymethylpterin monophosphate (PtP) bound to *M.tb* DHPS (PDBID: 1EYE), shown in Fig 3 [26]. This structure was selected due to its high resolution and the presence of the native pterin substrate within the catalytic site, providing a biologically validated reference for defining the active pocket. The structure exhibits highest sequence homology of more than 95% with DHPS sequences from each of *S. aureus, P. aeruginosa, E. coli* and *B. subtilis.* Moreover, the sequence identity and sequence homology in the binding-pocket region are highly conserved, exhibiting 99% homology and an RMSD of less than 1 Å. The 1EYE structure has been frequently used in previous structure-based analyses of DHPS. Due to these strong similarities and the highly conserved nature of DHPS across various organisms, the high-quality structure of PDB ID: 1EYE was selected for further analysis, making it an appropriate and reliable template for this study. Selected binding poses were then re-scored with DeepAtom, an in-house deep learning model for estimating binding affinities [27].

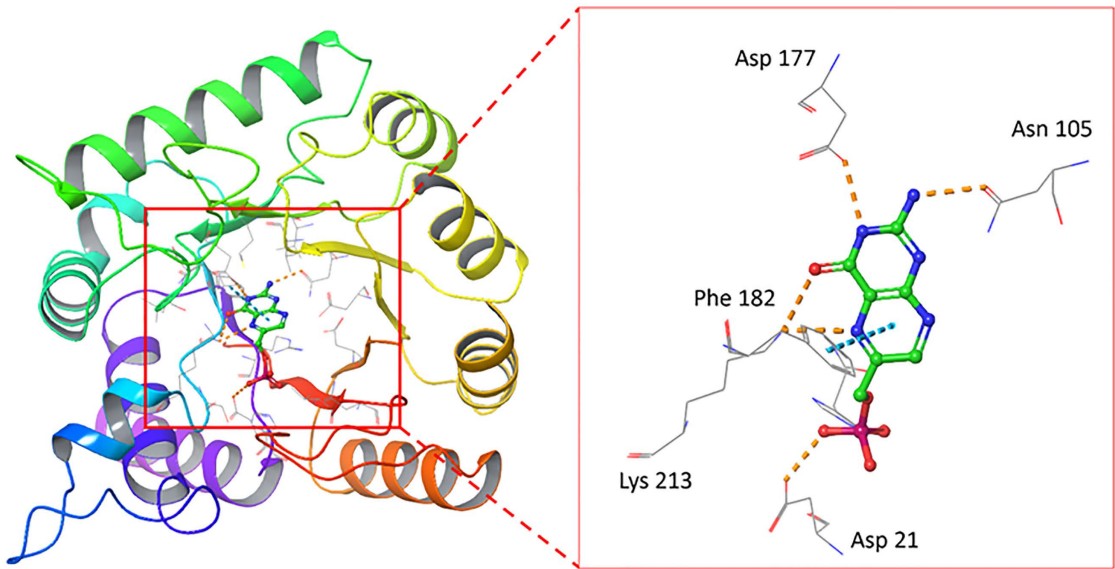

**Fig 3. Crystal structure of PtP bound to MtbDHPS from PDBID:1EYE, with a zoomed-in view of the binding site shown on the right.** Yellow dashed lines represent hydrogen bonding, and the blue line represents π-π interactions.

The sulfonamide analogues selected in this study were prioritized through a hierarchical computational workflow combined with practical considerations of structural relevance and accessibility. Initial screening shortlisted compounds with strong docking scores (≤ −8.5 kcal/mol) and favorable interactions with key DHPS residues such as Lys213, Asn105, and Phe182. A docking score cutoff of −8.5 kcal/mol was selected based on the empirical distribution of scores generated by QuickVina, which showed a natural inflection point near this value. This threshold is consistent with our previously validated docking workflow [28], where ligands scoring below −8.0 to −8.5 kcal/mol reproduced meaningful DHPS interactions. The cutoff was used only as a preliminary filter, after which compounds were prioritized based on binding mode inspection, DeepAtom rescoring, and MD stability. The sulfonamide scaffold had been previously optimized and synthesized in our laboratory, ensuring synthetic feasibility and alignment with known DHPS pharmacophoric elements. From this process, eight high-ranking candidates with favorable predicted binding modes were identified. However, among these eight priority candidates, only compounds 8802 and 7034 were commercially available in sufficient purity and quantity for biological testing at the time of the study. As the objective was to experimentally validate computationally predicted hits using readily obtainable molecules rather than undertake *de novo* synthesis of all candidates, these two accessible and computationally favorable compounds were selected for *in vitro* evaluation. Subsequent visual inspection, DeepAtom rescoring, and 100 ns MD simulations further supported the stability of these ligand–DHPS complexes (Fig 4, Table 1). The putative binding modes to DHPS can be seen in Fig 5 (8802) and Fig 6 (7034). Quinoline carboxamide 8802, which has previously demonstrated mild anticancer activity *in vitro* [29], engages in multiple favorable interactions with DHPS, including hydrogen bonding between its isoxazole oxygen and Asn105, hydrogen bonding between its amide carbonyl oxygen and Arg253, and cation-π interactions between its 2-phenyl substituent and Lys213. Hydrophobic interactions between its multiple aromatic rings and Val107, Met130, Leu180, and Phe182 were also observed. Sulfonamide 7034 captures a similar range of interactions with DHPS residues, including hydrogen bonding between its isoxazole oxygen

**8802**     **7034**

**Fig 4. Hit compounds selected from virtual screening.**

**Table 1. Binding scores for screening hits selected for biological validation. MM-GBSA values are relative to 7034.**

| Compound Code | Vina Score (kcal/mol) | DeepAtom Score (kcal/mol) | MM-GBSA $\Delta\Delta G_{bind}$ (kcal/mol) |
| --- | --- | --- | --- |
| 8802 | −9.0 | −10.3 | −7.6 |
| 7034 | −9.5 | −10.1 | 0 |
| LST-1 | −7.6 | n.d. | −8.1 |
| LST-2 | −7.4 | n.d. | +5.4 |

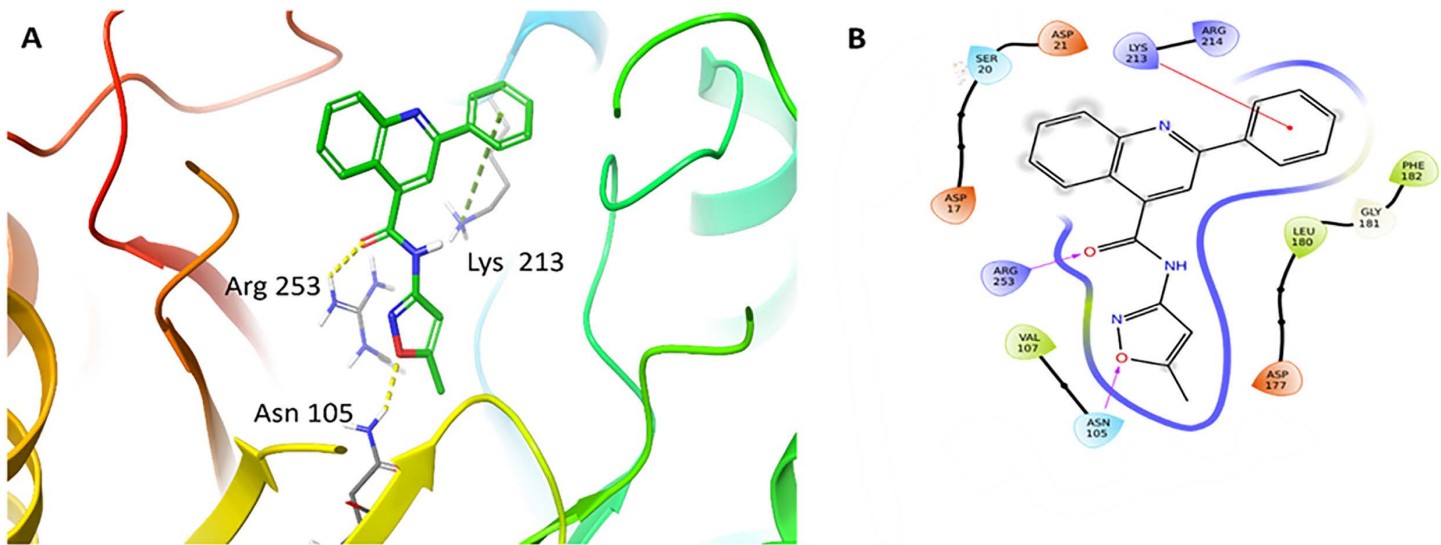

**Fig 5.  (A) Binding mode of compound 8802 docked to MtbDHPS, with key residues labeled. Yellow dotted lines represent hydrogen bonds, and the green dotted line represents cation-π interactions. (B) Interaction diagram of 8802 docked to MtbDHPS. Arrows in purple represent hydrogen bonds with donor at the base and acceptor at the arrowhead. Red lines show cation-π interactions.**

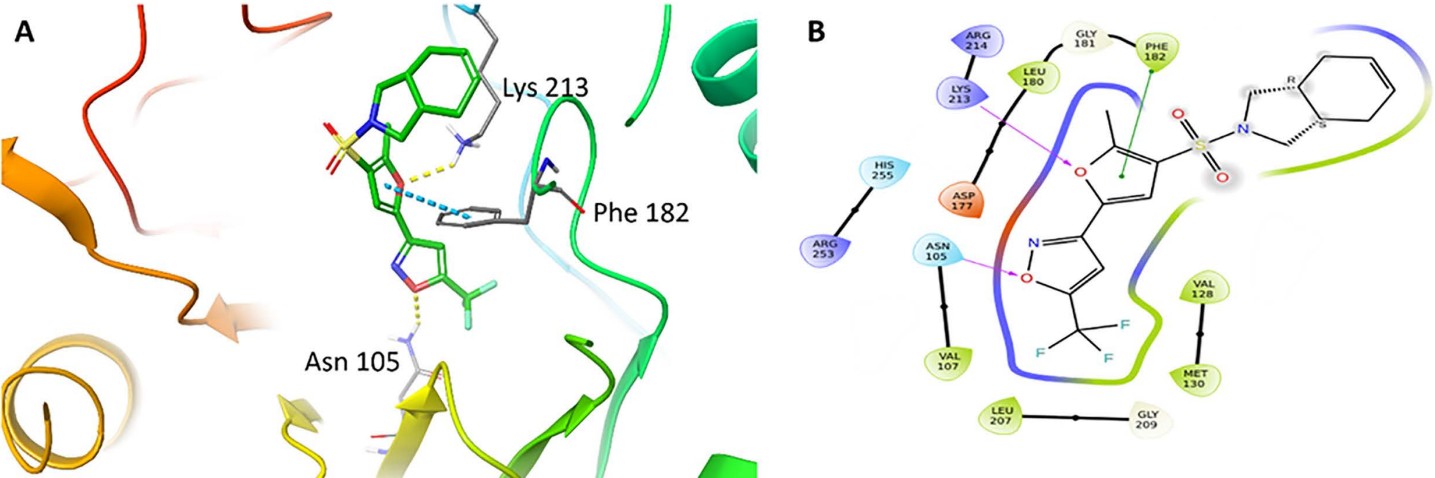

**Fig 6.  (A) Binding mode of compound 7034 docked to MtbDHPS, with key residues labeled. Yellow dotted lines represent hydrogen bonds and blue dotted line represents π-π interactions. (B) Interaction diagram of compound 7034 docked to MtbDHPS. Arrows in purple represent hydrogen bonds with donor at the base and acceptor at the arrowhead. The green line shows π-π interactions.**

and Asn105, hydrogen bonding between its furan oxygen and Lys213, and π-π interactions between its furan ring and Phe182. Similar beneficial hydrophobic interactions as with **8802** are also observed.

## Analogue development and docking

While the binding poses offered by the most promising *in silico* hits, **8802** and **7034**, were favorable, analogues of the former were also proposed and synthesized in an attempt to improve binding interactions with DHPS (Fig 7). It was

**Fig 7. Rationale for and Synthesis of Analogues of 8802.**

postulated that increasing the sp$^3$ character of the substrate via opening the isoxazole ring of **8802** into an analogous ethyl carboxamide or ethyl sulfonamide pendant would afford increased flexibility, therefore improving both solubility and the ability of the ligand to adopt more favorable binding conformations. Hydrogen bond donors in the form of said primary carboxamide or primary sulfonamide were also desirable for potentially capturing additional interactions with key residues of the DHPS pterin site. Analogues **LST-1** and **LST-2** were synthesized in moderate yield via HATU amide coupling from 2-phenylquinoline-4-carboxylic acid **12** and the appropriate primary amine.

Compounds **LST-1** and **LST-2** were also docked to DHPS (PDBID: 1EYE) using AutoDock Vina. Their predicted binding modes are shown in Fig 8 **(LST-1)** and Fig 9 **(LST-2)**. **LST-1** captures π-π interactions between its 2-phenyl moiety and Phe182 as well as cation-π interactions between the same 2-phenyl group and Lys213., The primary carboxamide oxygen of **LST-1** and sulfonamide oxygens of **LST-2** capture hydrogen bonding interactions with Lys213, though the latter also exhibits a water-mediated hydrogen bond with Asp177 (this water has been previously shown to form a bridged hydrogen bonding interaction between Asp177 and PtP [26]. **LST-2** is also predicted to capture multiple π-π interactions between its polyaromatic scaffold and Phe182 and Trp132.

## Molecular dynamics simulations studies

Rigid docking studies have several inherent limitations and can often produce false positive hits [22,30]. Since the binding of ligands to biological macromolecules occurs in a dynamic environment *in vivo*, further analysis of these results via molecular dynamics simulations is warranted to assess the quality of the proposed binding modes in greater detail [23,31]. To investigate the conformational stability of these docked compounds with the protein, the docked complexes were subjected to Molecular Dynamics (MD) Simulations with the AMBER software suite [32] (S12-S15 Figs).

The protein was assigned the ff19SB force field, while ligands were parameterized using GAFF2, with partial charges generated through the AM1-BCC method using Antechamber. Missing parameters were assigned using parmchk2. Protein–ligand complexes were solvated in an OPC truncated octahedron water box with a 10 Å buffer from the solute, using the solvateOct command in LEaP.

System neutralization was achieved by adding Na$^+$ and Cl$^-$ counterions, followed by additional ions to mimic 0.15 M physiological ionic strength using addIonsRand. The final topology and coordinate files were generated using LEaP.

All non-bonded interactions employed an 8.0 Å cutoff, while long-range electrostatics were treated using the Particle Mesh Ewald (PME) method. Energy minimization consisted of 5,000 steps of steepest descent followed by 5,000 steps of conjugate gradient minimization. The system was then gradually heated to 298 K under NVT ensemble conditions using Langevin dynamics (collision frequency γ = 1.0 ps$^{-1}$), followed by a 200 ps density equilibration under NPT conditions using the Monte-Carlo barostat.

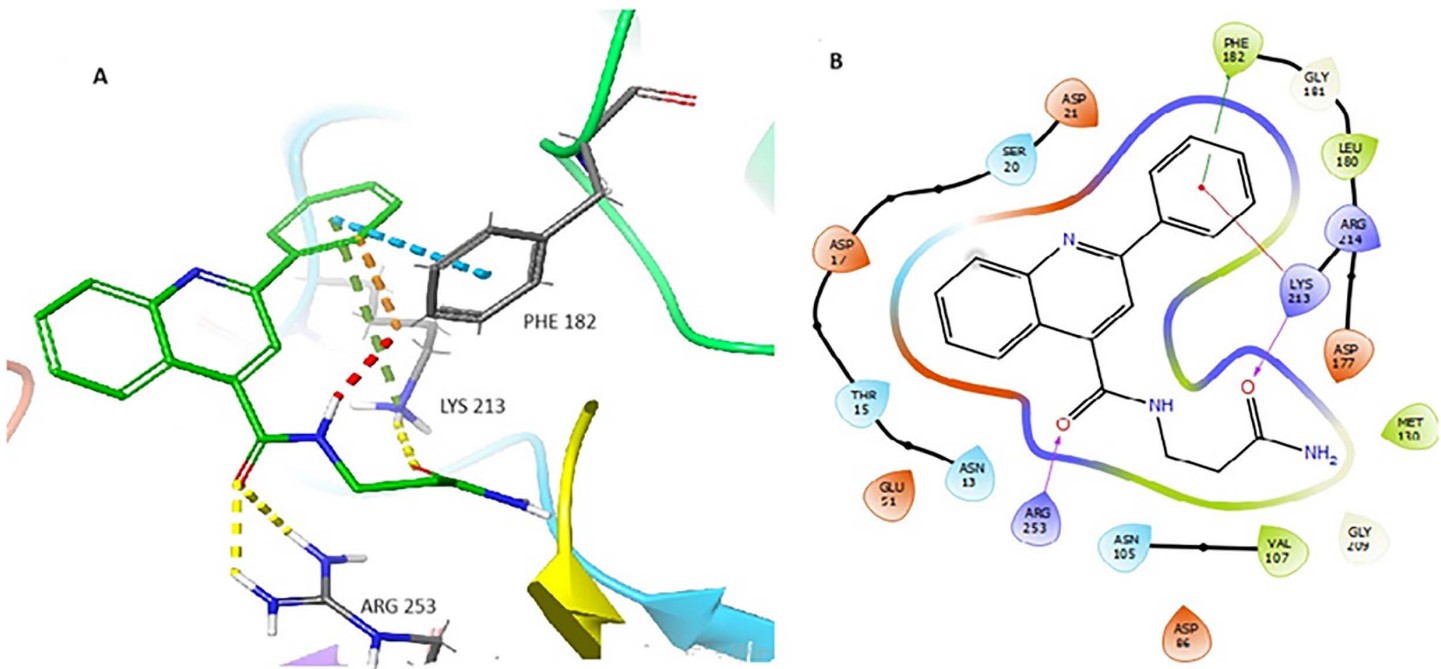

**Fig 8. (A)** Binding mode of LST-1 docked to MtbDHPS, with key residues indicated. Yellow dotted lines represent hydrogen bonds, blue dotted line represents π-π interactions, and green dotted lines show cation-π interactions. **(B)** Interaction diagram of LST-1 docked to MtbDHPS. Arrows in purple represent hydrogen bonds with the donor at the base and acceptor at the arrowhead. The green line indicates π-π interactions. Red lines show cation-π interactions.

Production simulations were run for 100 ns with a 2 fs timestep, employing SHAKE to constrain bonds involving hydrogens. Trajectory analyses, including RMSD and hydrogen bonding, were performed using cpptraj.

Compound **8802** is stable in the binding site with maximum RMSD (Root Mean Square Deviation) with respect to the docked pose around 2Å, as shown in S12 Fig. The higher RMSD values for the protein atoms with respect to the initial structure reflects the relaxation from crystal structure and adjustment to the presence of the ligand. Overall, this simulation suggests that, while **8802** remains relatively stable in its binding pose, the protein undergoes some structural adjustments over the course of the simulation.

Compound **7034** also displays a less stable RMSD, as shown in S13 Fig, indicating that the ligand remains in the pocket throughout the simulation, but is less tightly bound than **8802**. The protein undergoes similar rearrangements as in the case of **8802**. Stabilization after ~50 ns suggests that the protein reaches into a stable conformation after these initial fluctuations.

In the presence of **LST-1**, the protein quickly settles into a stable structure after ~20 ns. **LST-1** also stabilizes at an RMSD below 2.5 Å after the same time, indicating that the ligand reaches a favorable conformation different than the docked pose, and keeps its position within the DHPS binding site as shown in S14 Fig. The higher RMSD values obtained when considering all protein atoms (green line), as opposed to considering only the protein backbone (red line), suggest mostly side chain rearrangements, instead of large-scale movement.

The case of **LST-2**, as shown in S15 Fig, is very similar, with the protein reaching a stable conformation. The bound ligand also remains stable, with an RMSD around 1.5–2.5 Å after initial equilibration. This suggests the complex reaches equilibrium around 25 ns, with small fluctuations in both the protein and ligand during the remainder of the simulation.

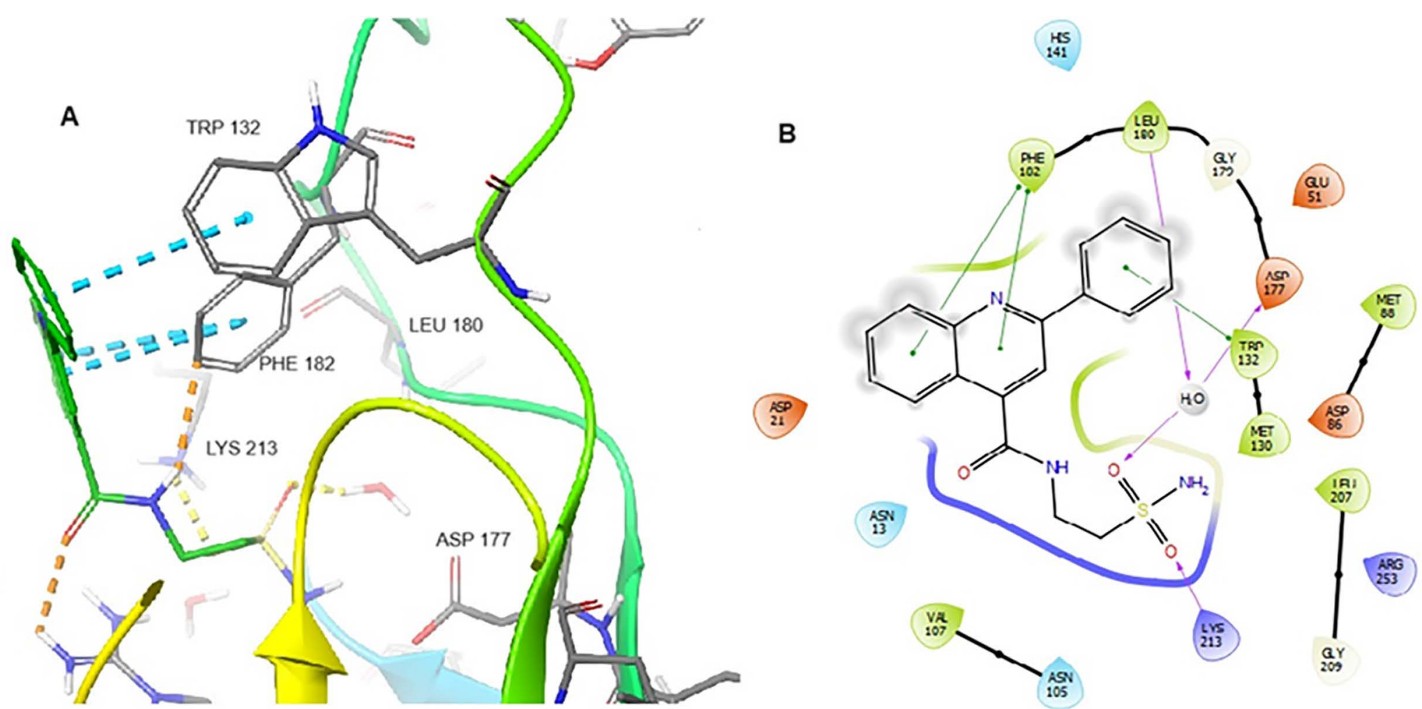

**Fig 9. (A) Binding mode of LST-2 (green) docked to MtbDHPS, with key residues indicated. Yellow dotted lines represent hydrogen bonds, and blue dotted lines represent π-π interactions. (B) Interaction diagram of LST-2 docked to MtbDHPS. Arrows in purple represent hydrogen bonds with the donor at the base and acceptor at the arrowhead. The green line shows π-π interactions.**

### MMGBSA binding free energy analysis

The binding free energy ($\Delta\Delta G$) for the docked complexes was estimated using the Molecular Mechanics with Generalized Born and Surface Area Solvation (MM-GBSA) method based on 5,000 simulation frames extracted from the simulated trajectories discussed above and utilizing two different solvent models. Both solvent models yield identical relative binding energies. While MM/GBSA methods are not suitable for calculating absolute binding energies, they are widely used for comparative analysis and relative binding energy estimations, which is beneficial for comparing the compounds presented in this study, taking as reference the compound with lowest initial docking score, **7034**. The results of these calculations are also shown in Table 1.

For both tested solvent models, compounds **8802** and **LST-1** exhibit a binding energy change of −7.6 and −8.1 relative to **7734**, respectively, suggesting that these should have more favorable binding to DHPS compared to compound **7034.** In contrast, **LST-2** binding energy was + 5.4 kcal/mol above that of **7034**, pointing to it being less effective.

### Prediction of physicochemical properties, drug-likeness, and ADMET properties

Since a significant percentage of clinical trial failures (~40−45%) occur due to unfavorable drug candidate Absorption, Distribution, Metabolism, Excretion, and Toxicity (ADMET) properties [33](Sun et al., 2022), consideration and optimization of ADMET properties in earlier stages of the drug discovery pipeline is an increasingly attractive means of mitigating risk. Due to the costly nature of the experimental determination of target compound ADMET properties, computational prediction has arisen as a valuable orthogonal approach due to its cost-effective and high-throughput nature [34] (Wu et al., 2020). To better evaluate the potential of these compounds as starting points for further development, their physico-chemical properties (Table) and ADMET profiles (Table 2) were predicted. *In silico* hit compounds **8802** and **7034** adhere

**Table 2. Predicted Physicochemical Properties and Drug-Likeness.**

| Compound | MolPSA (Å$^2$) (< 90 Å$^2$) | Lipinski's Parameters | | | | Drug-Like Score(> 0) |
|---|---|---|---|---|---|---|
| | | nHBA (< 10) | nHBD (< 5) | clogP (< 5) | MW (< 500) | |
| **8802** | 53.2 | 5 | 1 | 3.9 | 329.4 | +0.37 |
| **7034** | 62.0 | 6 | 0 | 4.0 | 402.4 | -0.37 |
| **LST-1** | 66.2 | 5 | 3 | 2.8 | 319.4 | +0.67 |
| **LST-2** | 84.3 | 6 | 3 | 2.5 | 355.4 | +0.46 |

nHBA = number of hydrogen bond acceptors; nHBD = number of hydrogen bond donors; cLogP = calculated logarithm of the octanol/water partition coefficient; M.W. = molecular weight; Drug-Like Score = parameter predicting overall drug-likeness of the compound. Typical cutoffs for druglike compounds are shown for each parameter.

to Lipinski's Rule of 5, as do synthetic analogues **LST-1** and **LST-2** [35]. Furthermore, all compounds exhibit a Molecular Polar Surface Area (MolPSA) less than 90 Å$^2$, suggesting favorable intestinal and blood-brain barrier permeability [36,37], and the combination of this with their favorable Lipinski parameters led to each of these compounds being flagged as drug-like (Drug-Like Score > 0). Only **7034** showed a negative Drug-Like score, which raises concerns about this compound.

With respect to ADMET properties (Table 3), **8802** and its analogues are predicted to form fewer metabolites than **7034**. While **LST-1** and **LST-2** are predicted to have poor Caco-2 permeability, these were predicted to have acceptable solubility plus Madin-Darby canine kidney (MDCK) cell permeability scores. All compounds were predicted to have desirable intestinal absorption. None of these compounds are expected to exhibit human ether-a-go-go related (hERG) gene potassium channel liabilities, though **LST-1** and **LST-2** are both predicted to have significantly weaker activity at this off-target compared to virtual screening hits **8802** and **7034**. Finally, none of these compounds are anticipated to act as substrates or inhibitors of P-glycoprotein (P-gP), a critical central nervous system transporter [38].

## Antibacterial activity

The computational results suggested that **8802** and **LST-1** could prove DHPS inhibitors better than **7034** and **LST-2**. With these compounds in hand, the Minimum Inhibitory Concentrations (MICs) against key bacterial species was assessed (Table 4). The compound **7034** only exhibited mild activity against *P. aeruginosa* (MIC: 128 µg/mL). Compound **8802**, however, displayed improved activity across all tested bacterial strains, in agreement with our MMGBSA results, which

**Table 3. ADMET Profile of Selected DHPS Virtual Screening Hits and Analogues.**

| Cmpd. | No. of Metab-olites (1-8) | Caco-2 Perm. (> 0.90) | hERG pIC$_{50}$ (≤6) | SASA (300.0–1000.0) | S+ MDCK (5-1130) | Human Intestinal Abs. (>80%) | P-gP Sub-strate? | P-gP Inhibi-tor? |
|---|---|---|---|---|---|---|---|---|
| **8802** | 1 | 1.14 | 5.17 | 634.3 | 675.4 | 100.0 | No | No |
| **7034** | 4 | 0.98 | 5.19 | 647.0 | 354.2 | 96.6 | No | No |
| **LST-1** | 2 | −4.76 | 0.45 | 605.6 | 121.0 | 99.8 | No | No |
| **LST-2** | 0 | −5.60 | 0.59 | 640.5 | 193.3 | 99.7 | No | No |

Abbreviations: Cmpd. = Compound; No. = Number; Caco2 Perm. = permeability of compound across a human colon carcinoma cell line monolayer; SASA = solvent accessible surface area; S+MDCK = solubility plus Madin-Darby canine kidney (MDCK) cell permeability; hERG pIC$_{50}$ = negative log of the half maximal inhibitory concentration against the human ether-a-go-go related gene potassium channel (here, the safety cutoff is pIC$_{50}$ = 6); For human intestinal absorption, 80% is used as a cutoff to indicate a highly-absorbed compound; P-gP = P-glycoprotein.

**Table 4. Minimum Inhibitory Concentrations of Compounds against Selected Bacterial Strains, in µg/mL.**

| Compound | S. aureus ATCC25923 | P. aeruginosa ATCC27853 | E. coli ATCC35401 | B. subtilis ATCC6633 |
|---|---|---|---|---|
| 8802 | 128 | 64 | 64 | 128 |
| 7034 | >256 | 128 | >256 | >256 |
| LST-1 | 256 | 256 | 256 | 256 |
| LST-2 | 256 | 256 | 256 | 256 |
| FQ5* | 32 | 16 | 16 | 16 |

*Data for **FQ5** is reproduced here from our previous work (Noor et al., 2025).

suggested that **8802** should have superior DHPS binding compared with **7034**. The compound **8802** exhibited an MIC of 64 µg/mL against both *P. aeruginosa* and *E. coli*. Analogues of **8802**, i.e., **LST-1** and **LST-2** exhibited diminished activity compared to the hit compound, with MICs of 256 µg/mL against all tested bacterial strains, but still superior to **7034**. Compound **FQ5**, the *N*-acetyl analogue of known mild antibacterial and *p*ABA-site-binding compound 4-sulfanilamidobenzoic acid [39], was tested alongside these compounds as a positive control and was discussed in greater detail in our previous work [28].

The use of FQ5 as a positive control is justified given its previously established DHPS inhibitory properties in our prior *in silico* and *in vitro* work, making it an appropriate internal benchmark.

### Biophysical ligand-DHPS binding studies

To confirm binding to the pterin site of *E. coli* DHPS (*Ec*DHPS), compounds **8802**, **7024**, **LST-1**, and **LST-2** were tested in a site-specific fluorescence polarization pterin probe displacement assay [40] (Fig 10). Among the compounds tested, only 7034 showed any probe displacement, and even then, the effect was minimal and observed only at high concentrations, indicating that these compounds do not bind the pterin-binding site of DHPS with sufficient affinity to effectively displace this high affinity probe.

To further confirm compound binding to *Ec*DHPS, SPR analysis was performed. Compounds **7034**, **8802**, **LST-1**, and **LST-2** all exhibited weak binding at the highest tested concentrations (50 µM for each), and full concentration curves could not be obtained, as testing at higher concentrations was limited by their solubility limits (see Supporting Information).

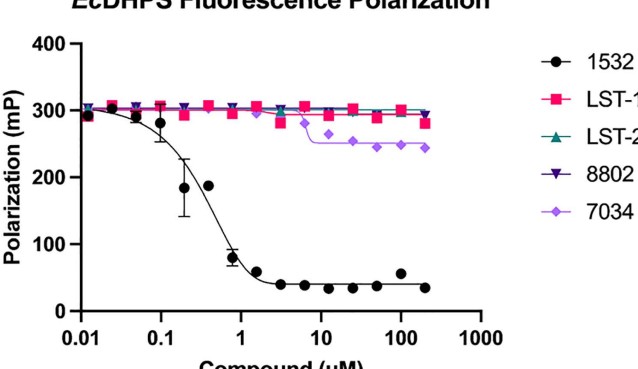

**Fig 10. Fluorescence Polarization Data.** Compound concentrations range from 0-200 µM. Data are an average of four replicates.

Compound 7034 binds the best among them, supporting the slight displacement seen in the fluorescence polarization data.

While docking, DeepAtom rescoring, and 100 ns MD simulations suggested that compounds 8802 and 7034 could form energetically favorable and stable interactions within the pterin pocket of DHPS, the fluorescence-polarization probe displacement assay and SPR measurements did not provide evidence of meaningful binding to the orthosteric pterin site of *E. coli* DHPS. This discrepancy likely arises from several factors. First, docking and MD are effective at ranking relative binding poses but do not reliably predict absolute binding affinities, particularly in systems where solvation effects, conformational flexibility, or entropic penalties dominate. Second, the computational modeling was performed using *M. tuberculosis* DHPS (PDB: 1EYE), whereas experimental assays used E. coli DHPS; subtle differences in loop regions adjacent to the pterin pocket may influence ligand entry or recognition. Third, MD simulations assess the stability of a ligand already placed inside an idealized, pre-organized crystal pocket, whereas FP and SPR require the ligand to physically enter and compete for the orthosteric site under dynamic solution conditions. Moreover the previous literature demonstrates difficulty in using SPR to generate full binding curves for compounds with mild antibacterial activity acting at this site [41].Together, these factors help explain why the MD complexes appear stable while experimental binding is weak. Consistent with the experimental data, we now interpret 8802 as a preliminary antibacterial lead whose mode of action is not attributable to classical pterin-site DHPS inhibition and may involve weak allosteric interactions or off-target mechanisms.

## Conclusion

In pursuit of novel pterin-site DHPS inhibitors to address the growing antimicrobial resistance crisis, a large virtual screen of more than 450,000 compounds led to the identification of hit compounds 8802 and 7034, along with two synthesized analogues (LST-1 and LST-2). Computational docking, DeepAtom rescoring, and molecular dynamics simulations initially suggested that these compounds could form stable interactions within the DHPS pterin pocket. *In silico* ADMET and physicochemical assessments further indicated a favorable overall profile for all four molecules.

In antibacterial testing, 7034 exhibited only mild activity against *P. aeruginosa*, whereas 8802, LST-1, and LST-2 showed modest but reproducible inhibitory activity against *P. aeruginosa, S. aureus, E. coli,* and *B. subtilis*, with 8802 emerging as the most potent consistent with computational predictions. However, fluorescence polarization and SPR studies revealed only weak binding to DHPS, suggesting that these compounds may not act through classical orthosteric pterin-site inhibition but may instead exert weak allosteric or off-target effects.

Overall, the combined computational and experimental findings position 8802 as a preliminary antibacterial lead with favorable predicted ADMET properties and *modest in vitro* activity. While the current derivatives display only modest antibacterial potential, they provide a promising starting point for further structural optimization. Future work involving rational design, targeted chemical modification, and expanded biological evaluation will be essential to improve potency, define the mechanism of action, and assess therapeutic relevance.

## Experimental procedures

### Protein preparation workflow

The 1.7 Å resolution x-ray crystal structure of DHPS from *M.tb* complexed with PtP was retrieved from the RCSB Protein Data Bank (PDBID: 1EYE) [26]. The protein was prepared for docking using Maestro Schrödinger (version 13.2) (Schrödinger, LLC) in a similar manner to our previously published procedure [28]: Missing residues and loop segments close to the active site were added using Prime. Hydrogen atoms were added after deleting any original ones and proper bond orders were assigned. In addition, water molecules were removed except for WAT 308, which coordinates with the $Mg^{2+}$ ion. PROPKA was used to sample hydrogen bonds while adjusting the orientations of the water molecules in the active site at pH 7.0 [42]. After that, the geometry of the protein-ligand complex was refined using OPLS4 force field

restrained minimization with convergence of heavy atoms to an RMSD of 0.3 Å [43](Lu et al., 2021). A PDBQT file for the processed protein was generated using AutoDock Tools [44].

### Ligand databases

*In silico* structure-based drug screening utilized chemical diversity libraries from commercial suppliers: TimTec ActiGlobe-50k (https://www.timtec.net, 50,000 compounds), Cambridge DIVERSet-CL (https://www.cambridgemedchem-consulting.com, 97,520 compounds), DrugBank 5.1.8 (https://go.drugbank.com, 11,172 compounds), and ChemDiv DivSet 300k (https://www.chemdiv.com/catalog/screening-libraries, 300,00 compounds). The ligands were obtained in SDF or SMILES format from the vendors and treated with Schrödinger's LigPrep to generate all possible protonation states at pH 7.4. Whenever chirality was not specified, all possible stereoisomers were generated up to a maximum of 32 stereoisomers per compound. Final structures were then converted to PDBQT format with OpenBabel v2.4.1 [45].

### Virtual screening procedure

Docking was performed with QuickVina 2.1 [25]. A 30x30x30Å grid box was centered on the 6-hydroxymethylpterin monophosphate ligand (x = 33.30, y = 4.29 and z = 39.35). Docking was performed using the prepared collection of compounds discussed in the previous section, with exhaustiveness set to 32. The results were then filtered to retain only ligands exhibiting binding energies of lower than approximately −8.5 kcal/mol. Cambridge, TimTec and DrugBank diversity libraries had 52, 59 and 18 hits respectively. The hits were further narrowed down through visual analysis of the binding mode, taking note of key hydrogen bonding, aromatic, and hydrophobic interactions, as well as proper ligand orientation within the pterin site. Selected binding poses were then rescored with DeepAtom, a Deep Learning model for estimation of binding affinities [27].

### Molecular dynamics simulations

The MD simulation protocols were divided into three main stages: initial preparation, which involved creating parameter files as inputs; a pre-processing phase; and the final simulation time phase. In the initial phase of file preparation, the Antechamber module of AMBER was used to generate parameter and coordinate files for each compound [46]. Subsequently, topology and coordinate files for the complex were created.

The protein was assigned the ff19SB force field, while ligands were parameterized using GAFF2, with partial charges generated through the AM1-BCC method using Antechamber. Missing parameters were assigned using parmchk2. Protein–ligand complexes were solvated in an OPC truncated octahedron water box with a 10 Å buffer from the solute, using the solvateOct command in LEaP. System neutralization was achieved by adding $Na^+$ and $Cl^-$ counterions, followed by additional ions to mimic 0.15 M physiological ionic strength using addIonsRand. The final topology and coordinate files were generated using LEaP [47–49].

The next steps for the protein system involved relaxing the solvent, heating, and equilibration to further prepare the system before the production run. Relaxation aims to create stable systems through a series of steps involving minimization, heating to a target temperature, and tapering off restraints on the solute. All non-bonded interactions employed an 8.0 Å cutoff, while long-range electrostatics were treated using the Particle Mesh Ewald (PME) method. The systems were first minimized to relax any close atomic contacts present. The maximum number of minimization cycles was set to 1000, and the system was equilibrated for 10 nanoseconds. The complexes were initially heated from 100 K to 298 K over 1 nanosecond at constant volume. Following this, the system was relaxed at constant pressure. Molecular Dynamics (MD) was run over 1 nanosecond at constant pressure and 298 K with a smaller restraint of 10 kcal/mol•Å² on the solute. The system was minimized with restraints only on the backbone of the protein. Then it was relaxed over 1 nanosecond at constant pressure with backbone restraints set to 10 kcal/mol•Å². Relaxation continued for 1 nanosecond at constant pressure with backbone restraints reduced to 1 kcal/mol•Å². Further relaxation for 1 nanosecond at constant pressure with backbone

restraints lowered to 0.1 kcal/mol•Å². The final relaxation step removed the restraints on the backbone, simulating for 1 nanosecond at constant pressure.

The production phase of the simulation was run using the same conditions as the final phase of equilibration to prevent an abrupt jump in potential energy due to a change in simulation conditions. The script was set up for a long-duration MD simulation of 50,000,000 MD steps, equivalent to 100 nanoseconds, under constant pressure and temperature conditions, suitable for studying the behavior of the system over time without any imposed restraints. The Visual Molecular Dynamics (VMD) tool was used for analyzing the MD simulation trajectories [50].

## Relative binding free energy (BFE) estimation

The molecular dynamics simulation trajectories were further used in an MM/GBSA binding free energy approach to calculate the interaction energy and solvation free energy for the complex, receptor and ligand and average the results to obtain an estimate of the binding free energy during the time course of dynamic environment [51].

For post-simulation analysis, the final MD trajectory was utilized for binding energy calculations. 100 frames were extracted from the 100 ns MD trajectory at regular intervals. This frame selection ensured an adequate representation of the system's conformational space while balancing computational efficiency. The MM-GBSA method was employed to evaluate the binding free energies, with entropy contributions calculated using normal mode analysis. Two solvation models were used: igb = 2 corresponds to the default solvation model; igb = 5 corresponds to a more sophisticated model for accounting for additional solvation effects. (Both yielded identical results, so only one result is shown in Table 1).

## Molecular properties and drug likeness

*In silico* prediction of Caco-2 permeability by pkCSM [52]; MolPSA, number of hydrogen bond acceptors and donors, molecular weight, and drug-likeness score using Molsoft [53] (MolSoft LLC); and ClogP, the number of metabolites, human intestinal absorption, solvent accessible surface area (SASA), and ADMET properties using ADMET Predictor® [54].

## Chemistry – general information

Chemical synthesis was conducted using standard techniques under air. Reagents and solvents were purchased from commercial chemical vendors and used without further purification. Reactions were monitored via thin layer chromatography. Automated flash column chromatography was performed using a Biotage Isolera One purification system. NMR spectra were obtained at room temperature using a Bruker Avance NEO-600 Spectrometer (¹H NMR: 600 MHz; ¹³C NMR: 151 MHz). All spectra were visualized using MestReNova 11.0, and all structures shown were drawn using ChemDraw 18.1. The solvent used for obtaining spectral data was DMSO-*d6* (¹H NMR: 2.50 ppm, ¹³C NMR: 39.52 ppm). All peaks were referenced either to the solvent peak or to TMS (¹H NMR: 0.00 ppm, ¹³C NMR: 0.00 ppm).

## Chemistry – synthesis

**N-(3-amino-3-oxopropyl)-2-phenylquinoline-4-carboxamide (LST-1).** To a vial was added 2-phenylquinoline-4-carboxylic acid (60.4 mg, 0.242 mmol, 1.0 eq.) and HATU (94.3 mg, 0.248 mmol, 1.0 eq.), which were dissolved in anhydrous DMF (2.5 mL). DIPEA (50 μL, 0.287 mmol, 1.2 eq.) was added, and the solution was stirred at room temperature for 6 minutes. 3-aminopropanamide (25.7 mg, 0.292 mmol, 1.2 eq.) was added, and the solution was stirred at room temperature overnight. The reaction was quenched with 10 mL AQ NaHCO₃ and extracted with multiple 20 mL portions of DCM. The organic layers were combined, dried over anhydrous Na₂SO₄, filtered, and concentrated. The crude material was purified via automated flash column chromatography (0–10% MeOH in DCM) to afford the title compound as an off-white solid (43.7 mg, 0.137 mmol, 57% yield). ¹H NMR (600 MHz, DMSO-*d₆*) δ 8.91 (t, *J* = 5.5 Hz, 1H), 8.32–8.28 (m, 2H), 8.20 (dd, *J* = 8.4, 0.7 Hz, 1H), 8.14–8.10 (m, 2H), 7.82 (ddd, *J* = 8.4, 6.9, 1.3 Hz, 1H), 7.63 (ddd, *J* = 8.3, 6.9, 1.1 Hz, 1H), 7.60–7.56 (m, 2H), 7.55–7.52 (m, 1H), 7.44 (s, 1H), 6.91 (s, 1H), 3.59–3.54 (m, 2H), 2.46 (t, *J* = 7.1 Hz, 2H). ¹³C

NMR (151 MHz, DMSO-$d_6$) δ 172.4, 166.6, 155.7, 147.9, 143.2, 138.2, 130.1, 129.9, 129.5, 128.9, 127.2, 127.1, 125.5, 123.4, 116.7, 36.0, 34.9.

**2-Phenyl-*N*-(2-sulfamoylethyl)quinoline-4-carboxamide (LST-2).** To a vial was added 2-phenylquinoline-4-carboxylic acid (59.6 mg, 0.239 mmol, 1.0 eq.) and HATU (91.6 mg, 0.241 mmol, 1.0 eq.), which were dissolved in anhydrous DMF (2.5 mL). DIPEA (50 µL, 0.287 mmol, 1.2 eq.) was added, and the solution was stirred at room temperature for 7 minutes. At this time, 2-aminoethane-1-sulfonamide (33.5 mg, 0.270 mmol, 1.1 eq.) was added, and the solution was stirred at room temperature overnight. The reaction was quenched with 10 mL AQ NaHCO$_3$ and extracted with multiple 20 mL portions of DCM. The organic layers were combined, dried over anhydrous Na$_2$SO$_4$, filtered, and concentrated. The crude material was twice purified via automated flash column chromatography (0–8% MeOH in DCM, then 50–100% EtOAc in Hexanes) to afford the title compound (52.8 mg, 0.149 mmol, 62% yield). $^1$H NMR (600 MHz, DMSO-$d_6$) δ 9.02 (t, *J* = 5.7 Hz, 1H), 8.33–8.29 (m, 2H), 8.24–8.22 (m, 1H), 8.21 (s, 1H), 8.13 (ddd, *J* = 8.5, 1.0, 0.5 Hz, 1H), 7.83 (ddd, *J* = 8.4, 6.9, 1.4 Hz, 1H), 7.65 (ddd, *J* = 8.4, 6.9, 1.3 Hz, 1H), 7.61–7.57 (m, 2H), 7.56–7.52 (m, 1H), 7.01 (s, 2H), 3.79–3.74 (m, 2H), 3.39–3.35 (m, 2H). $^{13}$C NMR (151 MHz, DMSO-$d_6$) δ 166.8, 155.7, 147.9, 142.7, 138.2, 130.2, 130.0, 129.5, 128.9, 127.23, 127.19, 125.4, 123.3, 116.9, 53.4, 34.7.

## Minimum inhibitory concentration (MIC) assays

MIC assays were performed in a similar manner as previously published [28]. MICs were determined following the Clinical and Laboratory Standards Institute (CLSI) guidelines for broth microdilution. All MIC values represent the modal MIC obtained from three independent biological replicates. Because MIC measurements rely on two-fold serial dilutions and produce ordinal rather than continuous data, formal statistical comparisons were not applied. Reproducibility was confirmed, with all replicate values falling within one dilution step. Each MIC plate included medium-only negative controls and CLSI-recommended quality-control strains to ensure assay validity. FQ5 was used as a positive control, as it is a sulfonamide derivative previously reported by the authors to show strong DHPS inhibitory activity in both *in silico* and *in vitro* studies These assays were performed using four standard strains of Gram-positive and Gram-negative bacteria: *Escherichia coli* ATCC 35401, *Pseudomonas aeruginosa* ATCC 27853, *Staphylococcus aureus* ATCC 25923, and *Bacillus subtilis* ATCC 6633. The MICs were determined using a micro-broth dilution method that followed the Clinical and Laboratory Standards Institute (CLSI) guidelines [55]. The three potential antibiotic compounds were solubilized in dimethyl sulfoxide and tested at concentrations that spanned the doubling dilution range from 256 µg/ml to 0 µg/ml. Mueller Hinton Broth was used for diluting the antimicrobial stocks. The overnight bacteria culture was diluted into 5 mL of fresh Tryptic Soy Broth and grown to an optical density (OD600) reaching one. Then the bacteria culture was diluted 100 times, two microliters of which was added to the well with 198 µL antimicrobial diluents in the 96-well plates. The MICs of bacteria were estimated after 24-hour growth with shaking (200 rpm) at 37°C.

## Fluorescence polarization assay

The fluorescence polarization assay protocol was adapted from the following reference [40]. 1 µM of *E. coli* DHPS was pre-incubated with 6 nM of probe in 40 mM HEPES pH 6.0, 4 mM MgCl$_2$ for 25 minutes. A labcyte Echo liquid handler was used to transfer each compound into a black polystyrene 384-well plate at doses ranging from 6 nM to 200 µM. The compounds were incubated with the probe-DHPS mixture for 10 minutes before reading the fluorescence. All FP measurements were performed using a BMG Labtech PHERAstar *FS* plate reader with excitation and emission wavelengths of 485 and 520 respectively.

## Surface plasmon resonance

SPR experiments were conducted at 20°C using a Cytiva Biacore 1S+SPR. His-tagged *Ec*DHPS constructs were immobilized on a Biacore NTA sensor chip. The chip was primed with a 300s injection of 350 mM EDTA, washed with running buffer (10 mM HEPES pH 7.4, 150 mM NaCl, 0.005% Tween20) and was charged with a 200s injection of 0.5 mM NiCl$_2$. The chip was washed

again and subject to activating using a 50:50 mixture of EDC and NHS. *E. coli* DHPS was injected until ~4000 RU of protein was captured. One flow cell on the chip was charged with $Ni^{2+}$ without adding protein to be used as a reference cell. The flow cells were washed again and deactivated with a 420s injection of ethanolamine. The flow cells were washed following deactivation.

Compounds were prepared in DMSO at concentrations ranging from 10 nM - 50 mM. 5 startups were completed prior to analyte injection. Each concentration of analyte was injected with a 120s contact time and 60s dissociation time. The flow cells were washed following each injection with analyte. A series of buffer-only (blank) injections and solvent corrections were included throughout the experiment to account for instrumental noise. The data was processed using the Cytiva evaluation software. Kinetic rate constants and affinities were determined by fitting the data to a 1:1 interaction model.

## Supporting information

**S1 Fig. Minimum Inhibitory Concentration (MIC), in µg/mL, of 8802 against selected bacterial strains.** (DOCX)

**S2 Fig. Minimum Inhibitory Concentration (MIC), in µg/mL, of 7034 against selected bacterial strains.** (DOCX)

**S3 Fig. Minimum Inhibitory Concentration (MIC), in µg/mL, of LST-1 against selected bacterial strains.** (DOCX)

**S4 Fig. Minimum Inhibitory Concentration (MIC), in µg/mL, of LST-2 against selected bacterial strains.** (DOCX)

**S5 Fig. Positive Control SPR Curve for Compound Lee1532 (Compound 10).** Concentration range: 10 nM – 50 µM. (DOCX)

**S6 Fig. SPR Results for Compound 8802.** Concentration range: 10 nM – 50 µM. RU ~ 3 at highest concentration. (DOCX)

**S7 Fig. SPR Results for Compound 7034.** Concentration range: 10 nM – 50 µM. RU ~ 5 at highest concentration. (DOCX)

**S8 Fig. SPR Results for LST-1.** Concentration range: 10 nM – 50 µM. RU = 1.25 at highest concentration. (DOCX)

**S9 Fig. SPR Results for LST-2.** Concentration range: 10 nM – 50 µM. RU ~ 2.5 at highest concentration. (DOCX)

**S10 Fig. $^1$H (400 MHz) and $^{13}$C NMR (151 MHz) spectra in DMSO-*d6* for LST-1.** (DOCX)

**S11 Fig. $^1$H (400 MHz) and $^{13}$C NMR (151 MHz) spectra in DMSO-*d6* for LST-2.** (DOCX)

**S12 Fig. RMSD of the 8802-MtbDHPS complex from 0–110 ns.** (DOCX)

**S13 Fig. RMSD of the 7034-MtbDHPS complex from 0–110 ns.** (DOCX)

**S14 Fig. RMSD of the LST-1-MtbDHPS complex from 0–110 ns.** (DOCX)

**S15 Fig. RMSD of the LST-2-MtbDHPS complex from 0–110 ns.**
(DOCX)

## Acknowledgments

The authors would like to thank COMSATS University Islamabad the International Islamic University Islamabad (IIUI) for granting a leave of study to Tahira Noor, which enabled the completion of this research. Finally, the authors would like to thank the Department of Medicinal Chemistry, College of Pharmacy, University of Florida (Gainesville, Florida, USA), for accommodating this research and the members of the Li lab for their steadfast support throughout this project.

## Author contributions

**Conceptualization:** Gustavo Seabra, Chenglong Li, Abdul Rauf Siddiqi.

**Data curation:** Tahira Noor, Daniel C. Schultz.

**Formal analysis:** Tahira Noor, Daniel C. Schultz, Chenglong Li.

**Investigation:** Tahira Noor, Daniel C. Schultz, Chenglong Li.

**Methodology:** Tahira Noor, Daniel C. Schultz, Gustavo Seabra, Chenglong Li.

**Project administration:** Abdul Rauf Siddiqi.

**Validation:** Yuting Zhai, Hannah E. Snoke, Suyeun Noh, Gustavo Seabra, Richard E. Lee, Kwangcheol Casey Jeong.

**Visualization:** Tahira Noor, Daniel C. Schultz, Chenglong Li.

**Writing – original draft:** Tahira Noor, Daniel C. Schultz.

**Writing – review & editing:** Tahira Noor, Yuting Zhai, Hannah E. Snoke, Suyeun Noh, Gustavo Seabra, Richard E. Lee, Kwangcheol Casey Jeong, Chenglong Li, Abdul Rauf Siddiqi.

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
