## [Decision Letter · Decision Letter 0]

2 Oct 2025

Dear Dr. Siddiqi,

Thank you for submitting your manuscript to PLOS ONE. After careful consideration, we feel that it has merit but does not fully meet PLOS ONE’s publication criteria as it currently stands. Therefore, we invite you to submit a revised version of the manuscript that addresses the points raised during the review process.

**ACADEMIC EDITOR: Please properly provide the reply to the Reviewers´ comments and improve the manuscript accordingly.**

We look forward to receiving your revised manuscript.

Kind regards,

Otávio Augusto Chaves

Academic Editor

PLOS ONE

Journal Requirements:

2.  We note that this submission includes NMR spectroscopy data. We would recommend that you include the following information in your methods section or as Supporting Information files:

1) The make/source of the NMR instrument used in your study, as well as the magnetic field strength. For each individual experiment, please also list: the nucleus being measured; the sample concentration; the solvent in which the sample is dissolved and if solvent signal suppression was used; the reference standard and the temperature.

2) A list of the chemical shifts for all compounds characterised by NMR spectroscopy, specifying, where relevant: the chemical shift (δ), the multiplicity and the coupling constants (in Hz), for the appropriate nuclei used for assignment.

3)The full integrated NMR spectrum, clearly labelled with the compound name and chemical structure.

We also strongly encourage authors to provide primary NMR data files, in particular for new compounds which have not been characterised in the existing literature. Authors should provide the acquisition data, FID files and processing parameters for each experiment, clearly labelled with the compound name and identifier, as well as a structure file for each provided dataset. See our list of recommended repositories here: https://journals.plos.org/plosone/s/recommended-repositories

3. Please note that PLOS One has specific guidelines on code sharing for submissions in which author-generated code underpins the findings in the manuscript. In these cases, we expect all author-generated code to be made available without restrictions upon publication of the work. Please review our guidelines at https://journals.plos.org/plosone/s/materials-and-software-sharing#loc-sharing-code and ensure that your code is shared in a way that follows best practice and facilitates reproducibility and reuse.

“The authors would like to thank COMSATS University Islamabad and FDP Grant # 17-5/FBSI[1]002/HEC/Sch-FDP/2018 of HEC, Pakistan, for providing financial support and resources for Ta[1]hira Noor to pursue a one-year secondment under the supervision of Dr. Chenglong Li at the Uni[1]versity of Florida. The authors would also like extend their gratitude to the International Islamic University Islamabad (IIUI) for granting a leave of study to Tahira Noor, which enabled the com[1]pletion of this research. Finally, the authors would like to thank the Department of Medicinal Chemistry, College of Pharmacy, University of Florida (Gainesville, Florida, USA), for accom[1]modating this research and the members of the Li lab for their steadfast support throughout this project.”

“Ms. Tahira Noor was awarded a one-year secondment under the supervision of Dr. Chenglong Li at the University of Florida, USA from Higher Education Commission (HEC)Pakistan. The details of grant number are given below.

FDP Grant # 17-5/FBSI-002/HEC/Sch-FDP/2018. The URL is https://www.hec.gov.pk/english/services/faculty/fdp/Pages/Financial-Benefits.aspx

HEC has played no role in the study design, data collection and analysis, decision to publish, or preparation of the manuscript.”

5. Please provide a complete Data Availability Statement in the submission form, ensuring you include all necessary access information or a reason for why you are unable to make your data freely accessible. If your research concerns only data provided within your submission, please write "All data are in the manuscript and/or supporting information files" as your Data Availability Statement.

6. Please ensure that you refer to Figure 13 in your text as, if accepted, production will need this reference to link the reader to the figure.

8. We are unable to open your Supporting Information file [1EYE_SI_2025-07-23_GMS-DCS.docx]. Please kindly revise as necessary and re-upload.

Reviewers' comments:

Reviewer's Responses to Questions

**Comments to the Author**

1. Is the manuscript technically sound, and do the data support the conclusions?

Reviewer #1: No

Reviewer #2: Partly

2. Has the statistical analysis been performed appropriately and rigorously?

Reviewer #1: No

Reviewer #2: No

3. Have the authors made all data underlying the findings in their manuscript fully available?

Reviewer #1: Yes

Reviewer #2: Yes

4. Is the manuscript presented in an intelligible fashion and written in standard English?

Reviewer #1: Yes

Reviewer #2: Yes

Reviewer #1: The manuscript is presented with toppic which is of current needs in anti microbial therapy. The work presented comprises the use of insilico methods to find out novel leads helpful in resitance. The below mentioned revesions are required from the authors to further strenthen the work.

1. The presented manuscript does not provide the enough information regarding the selected target protein. As, the work presented heavily depends on the virtual and InSilico methods, it becomes necessary that authors include technical discussion of the protein structure used. Critical details such as, the availability of other protein structure and the rational for selecting the specific entry should be clearly described. The availability of the alternative protein structure confirmation should be discussed. In absence of this information, it is difficult to assess the reliability of docking and simulation results, as protein structure quality and confirmational variability have direct impact on accuracy of computational predictions.

2. In the present work, the docking protocol used is lacking the validation through standard approach such as redocking through co-crystalized ligand and bench marking with known inhibitors. Without such validation, it is difficult to assess the reliability of docking poses and binding scores presented. It is recommended to add the redocking step with RMSD evaluation and cross validation with known active and inactive compounds will add more strength to the work.

3. The authors of the manuscript have not explained the rational for selecting -8.5kcal/mol as a cutoff docking score. Docking scores are relative and program dependent, and without justification it is difficult to understand why specific cutoff used to choose the hits.

4.MD protocol is written in detail but lacks certain important parameters for reproducibility of the results. Ligand parameterization, box dimension, ion neutralization and non-bounded cutoffs are not specified. It is further recommended to add the additional analysis such as hydrogen bonding, RMSF, RG would provide stronger evidence of ligand stability and protein confirmation.

Reviewer #2: The study integrates virtual screening, molecular dynamics, compound synthesis, and antibacterial evaluation, which is technically sound in design. However, the rationale for compound and PDB selection is weak, and there is a clear mismatch between computational predictions and experimental binding data. The MIC assays also lack detail on replicates and statistical rigor. The conclusions are in places overstated relative to the experimental findings.

The manuscript does not provide sufficient information on replicates, error analysis, or statistical methods for the MIC assays. Without this, the robustness of the biological findings cannot be confirmed.

The manuscript includes detailed computational protocols, docking data, predicted ADMET properties, and experimental MIC values. The data availability statement appears compliant with PLOS ONE requirements.

The manuscript is understandable and conveys the scientific content, but there are frequent grammatical issues and stylistic problems. Sentences are sometimes overly long, and subjective adverbs (interestingly, gratifyingly) should be avoided. Professional editing is recommended.

The manuscript addresses an important challenge in antibiotic discovery and identifies novel compounds with potential antibacterial activity. The integration of computational and experimental methods is a strength. However, several weaknesses need attention before the work is suitable for publication:

Compound selection: Only two screening hits were advanced, apparently based on docking scores and availability. Please clarify why other top-ranked compounds were not pursued, and whether diversity, novelty, or drug-likeness filters were considered.

Choice of structural template (PDB ID 1EYE): You used M. tuberculosis DHPS for docking, but tested compounds against E. coli, P. aeruginosa, S. aureus, and B. subtilis. Please justify this choice and discuss potential structural differences.

Computational vs. experimental contradiction: Docking/MD predicted strong binding, but fluorescence polarization and SPR assays indicated weak or no binding. This discrepancy must be addressed directly, rather than attributed only to “possible allosteric binding.”

Antibacterial assays: Provide details on the number of replicates and statistical treatment. The rationale for using FQ5 as a positive control should be expanded, and inclusion of standard sulfonamides or trimethoprim as controls is strongly advised.

Overstated conclusions: The discussion currently makes stronger claims than the data justify. Please temper the conclusions and present compound 8802 as a preliminary lead, not a confirmed pterin-site inhibitor.

Language and formatting: Revise the manuscript for grammar and clarity, ensure consistent reference style, and properly embed figures and tables.

Overallthe manuscript, it is promising but requires major revision to strengthen its rationale, methodological transparency, and balance between computational predictions and experimental evidence.

**Do you want your identity to be public for this peer review?** For information about this choice, including consent withdrawal, please see our Privacy Policy

Reviewer #1: **Yes:** sharav A desai

Reviewer #2: **Yes:** Mohammad Arman

---

## [Author Response · Author response to Decision Letter 1]

17 Dec 2025

we have attached the reviewer response in a separate attached document in which rebuttal for both reviewers has been submitted.

---

## [Decision Letter · Decision Letter 1]

13 Jan 2026

A Virtual Screening and Molecular Dynamics Approach in Search of Novel Antibiotic Chemotypes

PONE-D-25-47619R1

Dear Dr. Siddiqi,

We’re pleased to inform you that your manuscript has been judged scientifically suitable for publication and will be formally accepted for publication once it meets all outstanding technical requirements.

Kind regards,

Otávio Augusto Chaves

Academic Editor

PLOS One

Additional Editor Comments (optional):

Reviewers' comments:

Reviewer's Responses to Questions

**Comments to the Author**

Reviewer #1: All comments have been addressed

2. Is the manuscript technically sound, and do the data support the conclusions?

Reviewer #1: Yes

3. Has the statistical analysis been performed appropriately and rigorously?

Reviewer #1: Yes

4. Have the authors made all data underlying the findings in their manuscript fully available?

Reviewer #1: Yes

5. Is the manuscript presented in an intelligible fashion and written in standard English?

Reviewer #1: Yes

Reviewer #1: I have carefully reviewed your responses to the comments provided during the first round of review, along with the revised version of the manuscript.

I am pleased to note that all the comments raised earlier have been satisfactorily addressed and appropriately justified.

**Do you want your identity to be public for this peer review?** For information about this choice, including consent withdrawal, please see our Privacy Policy

Reviewer #1: **Yes:** Dr. Sharav A. Desai

---

## [Editor Report · Acceptance letter]

PONE-D-25-47619R1

PLOS One

Dear Dr. Siddiqi,

I'm pleased to inform you that your manuscript has been deemed suitable for publication in PLOS One. Congratulations! Your manuscript is now being handed over to our production team.

Kind regards,

on behalf of

Dr. Otávio Augusto Chaves

Academic Editor

PLOS One